# Adaptive Instance-wise Multi-view Clustering

## ABSTRACT

Multi-view clustering has garnered attention for its effectiveness in addressing heterogeneous data by unsupervisedly revealing underlying correlations between different views. As a mainstream method, multi-view graph clustering has attracted increasing attention in recent years. Despite its success, it still has some limitations. Notably, many methods construct the similarity graph without considering the local geometric structure and exploit coarse-grained complementary and consensus information from different views at the view level. To solve the shortcomings, we focus on local structure consistency and fine-grained representations across multiple views. Specifically, each view's local consistency similarity graph is obtained through the adaptive neighbor. Subsequently, the multi-view similarity tensor is rotated and sliced into fine-grained instance-wise slices. Finally, these slices are fused into the final similarity matrix. Consequently, cross-view consistency can be captured by exploring the intersections of multiple views in an instance-wise manner. We design a collaborative framework with the augmented Lagrangian method to refine all subtasks towards optimal solutions iteratively. Extensive experiments on several multi-view datasets confirm the significant enhancement in clustering accuracy achieved by our method.

## CCS CONCEPTS

• **Computing methodologies → Cluster analysis**.

## KEYWORDS

Machine Learning, Unsupervised Learning, Multi-view Clustering

## 1 INTRODUCTION

In recent years, with the rapid development of Information Technology, accessing and collecting data has become significantly more convenient through various means [26]. In the big data era, it is common to encounter situations where the collected data covers the information from multiple views [12]. For example, a patient's medical data may include physiological indicators, symptom descriptions, and treatment history. Each of them represents an aspect of the patient's health status, and there is a supplementary relationship between different views. These data are known as multi-view data, and each view contains partially independent and complementary information. In Figure 1, We listed several common multi-view data. Each view is sufficient for learning, and all views collectively

*ACM MM, 2024, Melbourne, Australia*
© 2024 Copyright held by the owner/author(s). Publication rights licensed to ACM.
ACM ISBN 978-x-xxxx-xxxx-x/YY/MM
https://doi.org/10.1145/nnnnnnn.nnnnnnn

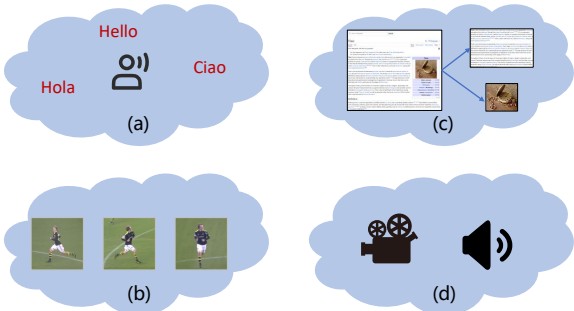

**Figure 1: Examples of multi-view data: (a) Documents with different languages, (b) Webpage with both textual and image data, (c) Photos of booter from different perspectives, (d) Multimedia with both video and audio signals.**

form a consensus latent representation [15]. Thus, integrating information from multiple views brings remarkable benefits towards unsupervised data clustering, and the underlying information embedded in the data can be fully utilized, improving the clustering quality. There have been a growing number of studies and applications dedicated to multi-view clustering (MVC) in recent years [2, 3], especially in real-world scenarios for anomaly detection in finance, data analysis in environmental science, and device collaboration in the Internet of Things.

Based on the technical mechanisms of the current MVC methods, they can be classified into two categories [3], namely heuristic-based MVC (HMVC) and neural networks-based MVC (NNMVC). Heuristic-based MVC extensively utilizes classical machine learning algorithms (Non-negative Matrix Factorisation, Graph Learning, Latent Representation Learning, and Tensor Learning) [8, 14]. Using prior knowledge and domain expertise, it uses heuristics to design clustering algorithms applicable to multi-view data. This approach emphasizes understanding data characteristics and problem context to guide algorithm design and parameter tuning for better adaptation to different application scenarios. Neural network-based MVC (NNMVC) is built on Deep Neural Networks (DNNs) with Deep Representation Learning or Deep Graph Learning [11, 31]. By creating complex neural network structures, the method can automatically learn high-level features and abstract representations in the data, effectively processing large-scale and high-dimensional multi-view data. With their notable nonlinear modeling capabilities, neural network-based MVC are well-equipped to identify complex relationships in data, offering valuable insights alongside traditional methodologies in complex scenarios.

Numerous multi-view clustering methods have demonstrated notable success in empirical studies [30]. However, we find some urgent shortcomings in the heuristic-based MVC graph learning method. First, current graph-based multi-view clustering algorithms highly rely on data similarity learning. Nevertheless, they generally lean towards subspace learning and analogous strategies, thereby overlooking the nuances of the data's local geometric structure.

Second, traditional methods usually fuse information from multiple views at the view level, which only considers coarse-grained information and reduces clustering accuracy.

As shown in Figure 2, we address the above issues simultaneously within a unified framework. As for the lack of local structure mining, we independently learn the similarity matrix of each view on local distance by assigning the adaptive and optimal neighbors. The local geometric structure directly implied by the data can be explored through the similarity propagation between neighbors. Regarding the fusion strategy, we obtain fine-grained information fusion in a self-weighted manner by instance-wise slicing of the multi-view similarity tensor. Finally, we introduce a unified model that learns both the adaptive similarity matrix and the instance-level structure of multi-view data to obtain better clustering results. In our collaborative model, we skillfully combine these two subtasks, alternatively driving each subtask toward the optimal solution with the augmented Lagrangian method. Experiments on several multi-view datasets demonstrate that our proposed approach significantly improves clustering accuracy.

## 2 RELATED WORK

clustering is the task of grouping a set of objects in such a way that objects in the same group (called a cluster) are more similar to each other than to those in other groups. In the era of big data, there are more and more multi-view data. Multi-view clustering has attracted increasing attention in recent years by aiming to exploit complementary and consensus information across multiple views.

In the past decade, multi-view graph clustering has been widely used as a mainstream method. However, traditional graph-based clustering methods have historically prioritized similarity construction, relying on a pre-determined data graph to segment data. These methods, however, introduce a significant dependency on the input affinity matrix, making clustering outcomes sensitive to the quality of the initial data representation. Recognizing this limitation, an innovative clustering method proposed by [19] introduces a transformative perspective. This method aims to enhance the acquisition of the data similarity matrix by dynamically assigning adaptive and optimal neighbors for each data point, considering local distance. A series of multi-view clustering methods have emerged in response to this paradigm shift. These approaches strive to address the challenges posed by traditional methods to improve the construction of the affinity matrix. Notable strategies include the automatic allocation of proper weights for each view [17], leveraging the graph smoothness assumption [6], devising a series of exponential functions for diverse scenarios [16], efficiently integrating graph learning with the fusion process [25], and concurrently harnessing graph information and embedding matrices [23]. These significant advancements depart from the conventional dependence on predetermined data graphs, ushering in a new era of more adaptive and sophisticated approaches to multi-view clustering. Moving away from rigid structures and embracing adaptability, these innovations empower clustering methods to respond dynamically to the intricacies and nuances inherent in diverse datasets, thereby enhancing multi-view clustering techniques' overall effectiveness and robustness.

It is crucial to highlight that the methods mentioned above aim to achieve a representation matrix with a primary focus on a view-oriented perspective. To delve into more specifics, it is evident that these methods undertake a sequential computation of one view at a time, subsequently consolidating their results at the view level. This approach, in turn, yields a coarse-grained view-level representation, wherein information integration occurs on a broader scale [21]. While effective for obtaining an overarching perspective and the complementary information from diverse modalities can easily be measured, this methodology may fail to capture the subtleties and intricacies within individual views. As a result, the representation matrix obtained may lack the granularity required for a more nuanced and detailed analysis of the underlying data structure. However, it's important to emphasize that the adaptive neighbor strategy scrutinizes the local connections inherent within the data. This signifies a meticulous examination of the intricate relationships existing in the local context of the dataset[27]. Consequently, the advantage offered by the adaptive neighbors strategy is susceptible to diminishing within a coarse-grained learning framework, where the nuanced details of local connections may not be adequately captured. Unlike the aforementioned MVC approaches, our proposed approach can grasp the local correlations among samples from various views and effectively eliminate redundant or erroneous information between these views.

## 3 METHOD

In this section, we first provide the notations used throughout the paper. Then, we derive the objective function of our method and present its optimization algorithm.

*Notation:* $\mathcal{X} = \left\{ \mathbf{X}^{(1)}, \ldots, \mathbf{X}^{(m)} \right\} \in \mathbb{R}^{d_v \times n}$ is a multi-view data, $d_v, n, m$ is the feature dimension of $\mathbf{X}^{(v)}$, number of samples, and number of views, respectively. $\mathcal{S} = \left\{ \mathbf{S}^{(1)}, \ldots, \mathbf{S}^{(m)} \right\} \in \mathbb{R}^{n \times n}$ is the similarity matrices generated from each view with adaptive neighbors. $\widetilde{\mathbf{S}} \in \mathbb{R}^{n \times n \times m}$ is a tensor composed of similarity matrices of all $m$ views, $\widetilde{\mathbf{S}}_i \in \mathbb{R}^{n \times m}$ is the $i$-th frontal slice of rotated $\widetilde{\mathbf{S}}$. $\mathbf{C} \in \mathbb{R}$ is the optimal fused similarity matrix.

### 3.1 Similarity Measurement

Similarity learning is a critical factor that determines the clustering result. Inspired by [19], the local connectivity of the data point can be mined from its neighbor data points, which helps improve the quality of the similarity matrix. Take the original data $\mathbf{X} = \{x_1, x_2, \ldots, x_n\}$ as an example. For simplicity, we employ Euclidean distance to learn the probabilistic $k$-nearest neighbors.

For a data point $x_i \in \mathbb{R}^{d_v}$, all $n$ data points in $\mathbf{X}$ can be regarded as connected neighbors with different probability $s$. The proximity of a data point to $x_i$ corresponds to a higher probability. Naturally, $\left\| x_i - x_j \right\|$ can be employed to gauge the Euclidean distance between two data points, and the optimal neighbors of $x_i$ can be determined by the following objective function:

$$\min_{(\mathbf{s}_i)^T \mathbf{1} = 1, 0 \leq \mathbf{s}_i \leq 1} \sum_{j=1}^{n} \left\| x_i - x_j \right\|_2^2 s_{ij}. \tag{1}$$

Eq. (1) has a straightforward solution, where only the nearest point has a probability of 1 while all others have 0. This leads to only

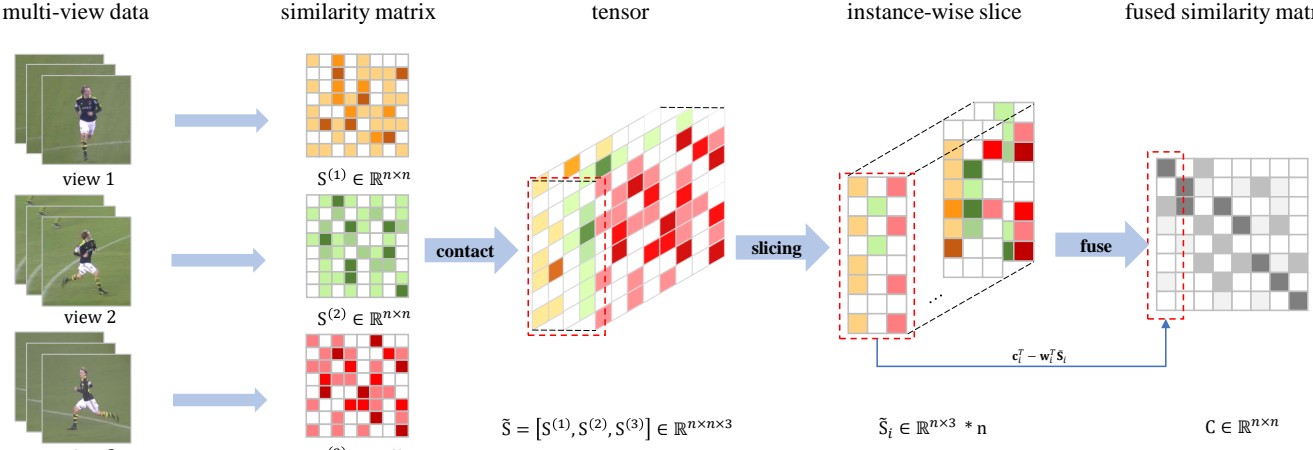

multi-view data  similarity matrix  tensor  instance-wise slice  fused similarity matrix

$S^{(1)} \in \mathbb{R}^{n \times n}$

$S^{(2)} \in \mathbb{R}^{n \times n}$

$S^{(3)} \in \mathbb{R}^{n \times n}$

**contact**  **slicing**  **fuse**

$\mathbf{c}_i^T - \mathbf{w}_i^T \bar{\mathbf{s}}_i$

$\widetilde{S} = [S^{(1)}, S^{(2)}, S^{(3)}] \in \mathbb{R}^{n \times n \times 3}$  $\widetilde{S}_i \in \mathbb{R}^{n \times 3} * n$  $C \in \mathbb{R}^{n \times n}$

**Figure 2: Framework of our method.**

one nearest neighbor of $x_i$ can be found. An alternative extreme solution is to disregard distance information completely:

$$\min_{(\mathbf{s}_i)^T \mathbf{1}=1, 0 \leq \mathbf{s}_i \leq 1} \sum_{j=1}^{n} s_{ij}^2, \qquad (2)$$

where all data points have same probability $\frac{1}{n}$.

Considering both of them, the optimal result should be balanced between Eq. (1) and Eq. (2):

$$\min_{(\mathbf{s}_i)^T \mathbf{1}=1, 0 \leq \mathbf{s}_i \leq 1} \sum_{j=1}^{n} \left( \|x_i - x_j\|_2^2 s_{ij} + \lambda s_{ij}^2 \right), \qquad (3)$$

where the second term is a regularization constraint. It is clear that Eq. (3) can be extended to multi-view domain:

$$\min_{\left(\mathbf{s}_i^{(v)}\right)^T \mathbf{1}=1, 0 \leq \mathbf{s}_i^{(v)} \leq 1} \sum_{v=1}^{m} \sum_{j=1}^{n} \left( \left\|x_i^{(v)} - x_j^{(v)}\right\|_2^2 s_{ij}^{(v)} + \lambda s_{ij}^{(v)2} \right), \quad (4)$$

The optimal neighbor assignment matrix $\mathbf{S}^{(v)}$ is precisely the similarity matrix encompassing the local geometric structure required for the multi-view clustering task.

### 3.2 Instance-Wise Fusion

Existing multi-view methods usually fuse multi-view complementary information in view-wise, which neglects the local correlation among samples from different views and disregards certain redundant or erroneous information between views.

We performed multi-view fusion at the instance level to enhance the fusion performance. This approach captures more local cross-view consistency and minimizes the impact of extra or inaccurate details. Refers to [4, 29], we perform a simple and effective transformation to achieve that. Firstly, the similarity matrices of multiple views $\{\mathbf{S}^{(v)} \in \mathbb{R}^{n \times n}\}_{v=1}^{m}$ are concatenated to a tensor $\widetilde{\mathbf{S}} \in \mathbb{R}^{n \times n \times m}$. Secondly, the tensor is rotated and sliced in the first dimension, and the $i$-th frontal slice $\widetilde{\mathbf{S}}_i \in \mathbb{R}^{n \times m}$ is the cross-view fusion of sample $i$. After obtaining $n$ slices, we perform instance-wise fusion to calculate the fused similarity matrix:

$$\min_{\mathbf{W}, \mathbf{C}} \alpha \sum_{i=1}^{n} \left\| \mathbf{c}_i^T - \mathbf{w}_i^T \widetilde{\mathbf{S}}_i \right\|_2^2 + \beta \|\mathbf{C}\|_F^2 \qquad (5)$$

$$\text{s.t. } \mathbf{C} \geq 0, \mathbf{w}_i \geq 0, \mathbf{C1} = \mathbf{1}, \mathbf{w}_i^T \mathbf{1} = 1.$$

$\mathbf{w}_i$ is the weight of $i$-th slice and $\mathbf{c}_i$ is the corresponding column in $C$ that integrates consistent information of sample $i$ from multiple views.

Lastly, combine Eq. (4) and Eq. (5) we formulate the objective function:

$$\min_{\mathbf{S}^{(v)}, \mathbf{W}, \mathbf{C}} \sum_{v=1}^{m} \sum_{i,j=1}^{n} \left\|x_i^{(v)} - x_j^{(v)}\right\|_2^2 s_{ij}^{(v)} + \frac{\lambda}{2} s_{ij}^{(v)2}$$

$$+ \frac{\alpha}{2} \sum_{i=1}^{n} \left\| \mathbf{c}_i^T - \mathbf{w}_i^T \widetilde{\mathbf{S}}_i \right\|_2^2 + \frac{\beta}{2} \|\mathbf{C}\|_F^2 \qquad (6)$$

$$\text{s.t. } \left(\mathbf{s}_i^{(v)}\right)^T \mathbf{1} = 1, 1 \geq \mathbf{s}_i^{(v)} \geq 0,$$

$$\mathbf{C} \geq 0, \mathbf{w}_i \geq 0, \mathbf{C1} = \mathbf{1}, \mathbf{w}_i^T \mathbf{1} = 1,$$

where $\mathbf{C} \in \mathbb{R}^{n \times n}$ is the fused simalirity matrix. A standard spectral clustering is executed with optimal $\mathbf{C}$ to obtain the final results.

## 4 OPTIMIZATION

As $\widetilde{\mathbf{S}}_i$ is closely related to $\mathbf{S}^{(v)}$ and can not be decomposed into elemental forms $\widetilde{\mathbf{s}}_{ij}$. With augmented Lagrangian method (ALM) [20], we introduce an intermediate variable $\mathbf{J}^{(v)}$ for each view:

$$\min_{\mathbf{S}^{(v)}, \mathbf{W}, \mathbf{C}} \sum_{v=1}^{m} \sum_{i,j=1}^{n} \left\|x_i^{(v)} - x_j^{(v)}\right\|_2^2 s_{ij}^{(v)} + \frac{\lambda}{2} s_{ij}^{(v)2}$$

$$+ \frac{\alpha}{2} \sum_{i=1}^{n} \left\| \mathbf{c}_i^T - \mathbf{w}_i^T \widetilde{\mathbf{J}}_i \right\|_2^2 + \frac{\beta}{2} \|\mathbf{C}\|_F^2 \qquad (7)$$

$$\text{s.t. } \left(\mathbf{s}_i^{(v)}\right)^T \mathbf{1} = 1, \mathbf{S}^{(v)} = \mathbf{J}^{(v)}, 1 \geq s_i^{(v)} \geq 0,$$

$$\mathbf{C} \geq 0, \mathbf{w}_i \geq 0, \mathbf{C1} = \mathbf{1}, \mathbf{w}_i^T \mathbf{1} = 1,$$

Hence, we can apply ALM and get the augmented Lagrangian function:

$$\min_{\mathbf{S}^{(v)}, \mathbf{W}, \mathbf{C}} \sum_{v=1}^{m} \sum_{i,j=1}^{n} \left\| x_i^{(v)} - x_j^{(v)} \right\|_2^2 s_{ij}^{(v)} + \frac{\lambda}{2} s_{ij}^{(v)^2}$$
$$+ \frac{\rho}{2} \sum_{v=1}^{m} \| \mathbf{S}^{(v)} - \mathbf{J}^{(v)} + \frac{1}{\rho} \mathbf{E}^{(v)} \|_F^2 \quad (8)$$
$$+ \frac{\alpha}{2} \sum_{i=1}^{n} \left\| \mathbf{c}_i^T - \mathbf{w}_i^T \widetilde{\mathbf{J}}_i \right\|_2^2 + \frac{\beta}{2} \| \mathbf{C} \|_F^2$$
$$\text{s.t.} \left( \mathbf{s}_i^{(v)} \right)^T \mathbf{1} = 1, 1 \geq s_i^{(v)} \geq 0,$$
$$\mathbf{C} \geq 0, \mathbf{w}_i \geq 0, \mathbf{C1} = \mathbf{1}, \mathbf{w}_i^T \mathbf{1} = 1,$$

where $\rho$ is penalty term and $\mathbf{E}^{(v)}$ is the Lagrangian multiplier.

[t] [1] Multi-view data $\mathbf{X}^{(v)}$ ($v = 1,2,...,m$), parameter $\alpha$ and $\beta$. Clustering result.
Update $\mathbf{S}^{(v)}$ according to Eq. (31). Update $\mathbf{J}^{(v)}$ by solving Eq. (17). Update $\mathbf{W}$ according to Eq. (20). Update $\mathbf{C}$ according to Eq. (23). Update Lagrangian multiplier $\mathbf{E}^{(v)}$ by:

$$\mathbf{E}_{t+1}^{(v)} = \mathbf{E}_t^{(v)} + \rho \left( \mathbf{S}^{(v)} - \mathbf{J}^{(v)} \right) \quad (9)$$

convergeConduct the standard spectral clustering on the optimal graph $\mathbf{C}$ to obtain the final clustering result.

## 4.1 Update $\mathbf{S}^{(v)}$:

According to Eq. (8), $\mathbf{S}^{(v)}$ is associated with the following function:

$$\min_{\mathbf{S}^{(v)}} \sum_{v=1}^{m} \sum_{i,j=1}^{n} \left\| x_i^{(v)} - x_j^{(v)} \right\|_2^2 s_{ij}^{(v)} + \frac{\lambda}{2} s_{ij}^{(v)^2}$$
$$+ \frac{\rho}{2} \sum_{v=1}^{m} \| \mathbf{S}^{(v)} - \mathbf{J}^{(v)} + \frac{1}{\rho} \mathbf{E}^{(v)} \|_F^2 \quad (10)$$
$$\text{s.t.} \left( \mathbf{s}_i^{(v)} \right)^T \mathbf{1} = 1, 1 \geq s_i^{(v)} \geq 0.$$

It is independent of different $i$ so that we can solve the following problem individually for each $i$, denote $\left\| x_i^{(v)} - x_j^{(v)} \right\|_2^2$ as $d_{ij}^{(v)}$, we have:

$$\min_{\mathbf{S}^{(v)}} \sum_{v=1}^{m} \sum_{j=1}^{n} d_{ij}^{(v)} s_{ij}^{(v)} + \frac{\lambda}{2} s_{ij}^{(v)^2}$$
$$+ \frac{\rho}{2} \sum_{v=1}^{m} \sum_{i,j=1}^{n} \left( s_{ij}^{(v)} + \frac{1}{\rho} e_{ij}^{(v)} - j_{ij}^{(v)} \right)^2 \quad (11)$$
$$\text{s.t.} \left( \mathbf{s}_i^{(v)} \right)^T \mathbf{1} = 1, 1 \geq s_i^{(v)} \geq 0.$$

denote $\frac{1}{\rho} e_{ij}^{(v)} - j_{ij}^{(v)}$ as $b_{ij}^{(v)}$, the formula is equivalent to:

$$\min_{\mathbf{S}^{(v)}} \sum_{v=1}^{m} \sum_{j=1}^{n} \left( \frac{\lambda + \rho}{2} \right) s_{ij}^{(v)^2} + \left( d_{ij}^{(v)} + \rho b_{ij}^{(v)} \right) s_{ij}^{(v)}$$
$$\text{s.t.} \left( \mathbf{s}_i^{(v)} \right)^T \mathbf{1} = 1, 1 \geq s_i^{(v)} \geq 0. \quad (12)$$

denote $d_{ij}^{(v)} + \rho b_{ij}^{(v)}$ as $f_{ij}^{(v)}$, the problem can be written in vector form as:

$$\min_{\mathbf{s}_i^T \mathbf{1} = 1, 0 \leq \mathbf{s}_i \leq 1} \sum_{v=1}^{m} \left\| \mathbf{s}_i^{(v)} + \frac{1}{\lambda + \rho} \mathbf{f}_i^{(v)} \right\|_2^2. \quad (13)$$

Note that $\mathbf{S}^{(v)}$ for each view is independent. Hence we can update $\mathbf{S}^{(v)}$ one by one,

$$\min_{\mathbf{s}_i^T \mathbf{1} = 1, 0 \leq \mathbf{s}_i \leq 1} \left\| \mathbf{s}_i^{(v)} + \frac{1}{\lambda + \rho} \mathbf{f}_i^{(v)} \right\|_2^2. \quad (14)$$

The solution to Eq. (14) will be detailed later.

## 4.2 Update $\mathbf{J}^{(v)}$:

When the other variables are fixed, Eq. (8) become:

$$\min_{\mathbf{J}^{(v)}} \frac{\rho}{2} \sum_{v=1}^{m} \| \mathbf{S}^{(v)} - \mathbf{J}^{(v)} + \frac{1}{\rho} \mathbf{E}^{(v)} \|_F^2 +$$
$$\frac{\alpha}{2} \sum_{i=1}^{n} \left\| \mathbf{c}_i^T - \mathbf{w}_i^T \widetilde{\mathbf{J}}_i \right\|_2^2, \quad (15)$$

then transform it into instance-wise form:

$$\min_{\mathbf{J}^{(v)}} \frac{\rho}{2} \sum_{i=1}^{n} \| \widetilde{\mathbf{S}}_i - \widetilde{\mathbf{J}}_i + \frac{1}{\rho} \widetilde{\mathbf{E}}_i \|_F^2 + \frac{\alpha}{2} \left\| \mathbf{c}_i^T - \mathbf{w}_i^T \widetilde{\mathbf{J}}_i \right\|_2^2. \quad (16)$$

By solving the first derivative of $\widetilde{\mathbf{J}}_i$, we have:

$$\widetilde{\mathbf{J}}_i = \left( \alpha \mathbf{w}_i \mathbf{w}^T + \rho \mathbf{I} \right)^{-1} \left( \alpha \mathbf{w}_i \mathbf{c}_i^T + \rho \widetilde{\mathbf{S}}_i + \widetilde{\mathbf{E}}_i \right) \quad (17)$$

## 4.3 Update $\mathbf{W}$:

The part of Eq. (8) with respect to $\mathbf{W}$ is:

$$\min_{\mathbf{W}} \frac{\alpha}{2} \sum_{i=1}^{n} \left\| \mathbf{c}_i^T - \mathbf{w}_i^T \widetilde{\mathbf{J}}_i \right\|_2^2$$
$$\text{s.t.} \mathbf{w}_i \geq 0, \mathbf{w}_i^T \mathbf{1} = 1. \quad (18)$$

Denote $\mathbf{1}\mathbf{c}_i^T - \widetilde{\mathbf{J}}_i \in \mathbb{R}^{m \times n}$ as $\mathbf{A}_i$, and simplify the above function

$$\min_{\mathbf{W}} \left\| \mathbf{w}_i^T \mathbf{A}_i \right\|_2^2$$
$$\text{s.t.} \mathbf{w}_i \geq 0, \mathbf{w}_i^T \mathbf{1} = 1, \quad (19)$$

Similarly, extreme values can be calculated using the derivation:

$$\mathbf{w}_i = \frac{\left( \mathbf{A}_i \mathbf{A}_i^T \right)^{-1} \mathbf{1}}{\mathbf{1}^T \left( \mathbf{A}_i \mathbf{A}_i^T \right)^{-1} \mathbf{1}}. \quad (20)$$

## 4.4 Update $\mathbf{C}$:

For $\mathbf{C}$ we need to optimize:

$$\min_{\mathbf{C}} \frac{\alpha}{2} \sum_{i=1}^{n} \left\| \mathbf{c}_i^T - \mathbf{w}_i^T \widetilde{\mathbf{J}}_i \right\|_2^2 + \frac{\beta}{2} \| \mathbf{C} \|_F^2$$
$$\text{s.t.} \mathbf{C} \geq 0, \mathbf{C1} = \mathbf{1}. \quad (21)$$

Reformulate it in a vector form, we have:

$$\min_{\mathbf{c}_i} (\alpha + \beta) \mathbf{c}_i \mathbf{c}_i^T - 2\alpha \mathbf{w}_i^T \widetilde{\mathbf{J}}_i \mathbf{c}_i$$
$$\text{s.t.} \mathbf{c}_i \geq 0, \mathbf{c}_i \mathbf{1} = 1. \quad (22)$$

Based on Eq. (22), we get the following compact formulation

$$\min_{\mathbf{c}_i \mathbf{1}=1, \mathbf{c}_i \geq 0} \left\| \mathbf{c}_i - \frac{\alpha}{\alpha + \beta} \mathbf{w}_i^T \widetilde{\mathbf{J}}_i \right\|_2^2, \tag{23}$$

We can follow the same process in solving Eq. (14) to solve Eq. (23).

## 4.5 Solution to Eq. (14)

The Lagrangian function of Eq. (14) is

$$\mathcal{L}\left(\mathbf{s}_i^{(v)}, \psi, \xi\right) = \frac{1}{2} \left\| \mathbf{s}_i^{(v)} + \frac{1}{\lambda + \rho} \mathbf{f}_i^{(v)} \right\|_2^2 \\ - \psi \left( \left(\mathbf{s}_i^{(v)}\right)^T \mathbf{1} - 1 \right) - \xi^T \mathbf{s}_i^{(v)}, \tag{24}$$

where $\psi$ and $\xi$ denote the Lagrange multipliers for the corresponding constraints.

Setting $\frac{\partial \mathcal{L}}{\partial \mathbf{s}_i^{(v)}} = 0$ and according to the KKT condition [1] $s_{ij}\xi_j = 0$, we have the following solution (denoted as $\widehat{s}_{ij}^{(v)}$) for $s_{ij}$

$$\widehat{s}_{ij}^{(v)} = \left( -\frac{1}{\lambda + \rho} f_{ij}^{(v)} + \psi \right)_+. \tag{25}$$

Without loss of generality, we order $f_{i1}^{(v)}, \ldots, f_{in}^{(v)}$ from small to large. If we constrain $s_{ij}$ having $k$ nonzero entries, we have:

$$-\frac{1}{\lambda + \rho} f_{ik}^{(v)} + \psi > 0, and -\frac{1}{\lambda + \rho} f_{i,k+1}^{(v)} + \psi \leq 0. \tag{26}$$

Combining Eq. (25) and the constraint $\left(\mathbf{s}_i^{(v)}\right)^T \mathbf{1} = 1$ we have

$$\psi = \frac{1}{k}\left( 1 + \frac{1}{\lambda + \rho} \sum_j^k f_{ik}^{(v)} \right), \tag{27}$$

, substituting Eq. (27) into Eq. (26),

$$k f_{ik}^{(v)} - \sum_j^k f_{ik}^{(v)} - \rho < \lambda \leq k f_{i,k+1}^{(v)} - \sum_j^k f_{ik}^{(v)} - \rho. \tag{28}$$

In order to constrain $\mathbf{s}_i^{(v)}$ to have exact $k$ nonzero elements, $\lambda$ can be set to

$$\lambda = k f_{i,k+1}^{(v)} - \sum_j^k f_{ik}^{(v)} - \rho. \tag{29}$$

Thus the overall optimal $\lambda$ (denoted as $\lambda^*$) of a given dataset is

$$\lambda^* = \frac{1}{n} \sum_{i=1}^n \left( k f_{i,k+1}^{(v)} - \sum_j^k f_{ik}^{(v)} - \rho \right). \tag{30}$$

According to Eq. (26), Eq. (27), and Eq. (29), the final solution for $s_{ij}^{(v)}$ is

$$s_{ij}^v = \begin{cases} \frac{f_{i,k+1}^{(v)} - f_{ij}^{(v)}}{k f_{i,k+1}^{(v)} - \sum_{h=1}^k f_{ih}^{(v)}} & j \leq k, \\ 0 & j > k. \end{cases} \tag{31}$$

## 5 EXPERIMENT

In this section, we compare our methodology with several state-of-the-art multi-view clustering on benchmark datasets. The empirical results show that our model achieves competitive results.

**Table 1: The clustering results on BBC dataset (%)**

| Method | ACC | NMI | Purity | F-score |
|---|---|---|---|---|
| Co-reg | 61.85 ± 1.11 | 55.29 ± 1.26 | 76.69 ± 0.96 | 60.10 ± 1.30 |
| Co-train | 64.82 ± 0.21 | 60.88 ± 0.02 | 78.88 ± 0.07 | 63.84 ± 0.05 |
| MLAN | 83.50 ± 0.00 | 66.03 ± 0.00 | 83.50 ± 0.00 | 73.75 ± 0.00 |
| AWP | 62.92 ± 0.00 | 42.34 ± 0.00 | 63.50 ± 0.00 | 49.99 ± 0.00 |
| MCGC | 33.43 ± 0.00 | 0.98 ± 0.00 | 33.87 ± 0.00 | 37.83 ± 0.00 |
| mPAC | 68.18 ± 0.00 | 47.42 ± 0.00 | 68.18 ± 0.00 | 63.45 ± 0.00 |
| CGD | 88.32 ± 0.21 | 71.49 ± 0.33 | 88.32 ± 0.21 | 82.07 ± 0.41 |
| FPMVS | 32.26 ± 0.00 | 2.91 ± 0.00 | 37.37 ± 0.00 | 27.59 ± 0.00 |
| LMVSC | 54.45 ± 2.60 | 37.58 ± 2.77 | 63.60 ± 0.07 | 44.37 ± 1.21 |
| CoMSC | 89.78 ± 0.00 | 74.29 ± 0.00 | 89.78 ± 0.00 | 84.16 ± 0.00 |
| COMVSC | 54.89 ± 0.00 | 28.31 ± 0.00 | 55.47 ± 0.00 | 49.40 ± 0.00 |
| MMGC | 50.51 ± 0.00 | 38.65 ± 0.00 | 60.00 ± 0.00 | 45.44 ± 0.00 |
| SLMVGC | 51.97 ± 0.00 | 32.41 ± 0.00 | 54.89 ± 0.00 | 39.64 ± 0.00 |
| Ours | **91.53 ± 0.00** | **77.33 ± 0.05** | **91.53 ± 0.00** | **85.82 ± 0.00** |

**Table 2: The clustering results on BBCSport dataset (%)**

| Method | ACC | NMI | Purity | F-score |
|---|---|---|---|---|
| Co-reg | 77.45 ± 1.46 | 54.87 ± 0.90 | 77.45 ± 1.46 | 67.77 ± 0.52 |
| Co-train | 75.43 ± 6.80 | 60.26 ± 1.85 | 78.37 ± 2.64 | 67.57 ± 3.29 |
| MLAN | 87.50 ± 0.00 | 76.52 ± 0.00 | 87.50 ± 0.00 | 84.27 ± 0.00 |
| AWP | 59.74 ± 0.00 | 43.06 ± 0.00 | 66.54 ± 0.00 | 47.42 ± 0.00 |
| MCGC | 37.13 ± 0.00 | 2.00 ± 0.00 | 37.13 ± 0.00 | 38.70 ± 0.00 |
| mPAC | 64.34 ± 0.00 | 42.42 ± 0.00 | 64.34 ± 0.00 | 57.31 ± 0.00 |
| CGD | 61.03 ± 0.00 | 49.47 ± 0.00 | 64.89 ± 0.00 | 59.07 ± 0.00 |
| FPMVS | 42.10 ± 0.00 | 14.78 ± 0.00 | 51.84 ± 0.00 | 32.74 ± 0.00 |
| LMVSC | 62.81 ± 6.46 | 43.15 ± 2.80 | 74.33 ± 6.42 | 48.83 ± 5.11 |
| CoMSC | 86.58 ± 0.00 | 74.98 ± 0.00 | 89.89 ± 0.00 | **83.69 ± 0.00** |
| COMVSC | 59.93 ± 0.00 | 30.00 ± 0.00 | 62.68 ± 0.00 | 49.50 ± 0.00 |
| MMGC | 71.32 ± 0.00 | 51.36 ± 0.00 | 73.90 ± 0.00 | 56.10 ± 0.00 |
| SLMVGC | 77.57 ± 0.00 | 60.73 ± 0.18 | 77.57 ± 0.00 | 63.72 ± 0.03 |
| Ours | **90.44 ± 0.00** | **77.84 ± 0.00** | **90.44 ± 0.00** | 81.44 ± 0.00 |

**Table 3: The clustering results on BBC12 dataset (%)**

| Method | ACC | NMI | Purity | F-score |
|---|---|---|---|---|
| Co-reg | 73.22 ± 5.03 | 53.57 ± 1.84 | 73.84 ± 4.17 | 64.25 ± 5.16 |
| Co-train | 74.51 ± 7.33 | 60.16 ± 0.65 | 79.35 ± 0.61 | 66.77 ± 3.73 |
| MLAN | 87.50 ± 0.00 | 76.52 ± 0.00 | 87.50 ± 0.00 | 84.27 ± 0.00 |
| AWP | 59.01 ± 0.00 | 42.56 ± 0.00 | 65.99 ± 0.00 | 46.46 ± 0.00 |
| MCGC | 37.13 ± 0.00 | 2.00 ± 0.00 | 37.13 ± 0.00 | 38.70 ± 0.00 |
| mPAC | 64.52 ± 0.00 | 43.82 ± 0.00 | 64.52 ± 0.00 | 57.70 ± 0.00 |
| CGD | 61.03 ± 0.00 | 49.47 ± 0.00 | 64.89 ± 0.00 | 59.07 ± 0.00 |
| FPMVS | 42.10 ± 0.00 | 14.78 ± 0.00 | 51.84 ± 0.00 | 32.74 ± 0.00 |
| LMVSC | 62.62 ± 6.81 | 45.23 ± 2.49 | 72.73 ± 7.18 | 48.91 ± 5.91 |
| CoMSC | 86.58 ± 0.00 | 74.98 ± 0.00 | 89.89 ± 0.00 | 83.69 ± 0.00 |
| COMVSC | 59.93 ± 0.00 | 30.00 ± 0.00 | 62.68 ± 0.00 | 49.50 ± 0.00 |
| MMGC | 71.32 ± 0.00 | 51.36 ± 0.00 | 73.90 ± 0.00 | 56.10 ± 0.00 |
| SLMVGC | 77.57 ± 0.00 | 60.52 ± 0.00 | 77.57 ± 0.00 | 63.69 ± 0.00 |
| Ours | **94.85 ± 0.00** | **84.53 ± 0.00** | **94.85 ± 0.00** | **90.10 ± 0.00** |

## 5.1 Experiment Setup

Five multi-view datasets are used in our experiment: **BBC**, **BBCSport**, **BBC12**, **HW2**, **ORL**, and **cora**.

**Table 4: The clustering results on HW2 dataset (%)**

| Method | ACC | NMI | Purity | F-score |
|--------|-----|-----|--------|---------|
| Co-reg | 55.28 ± 1.84 | 46.89 ± 1.04 | 58.13 ± 2.15 | 42.34 ± 0.60 |
| Co-train | 82.28 ± 4.87 | 72.12 ± 3.36 | 82.68 ± 4.30 | 71.30 ± 4.48 |
| MLAN | 81.07 ± 0.02 | 83.89 ± 0.04 | 85.52 ± 0.02 | 79.68 ± 0.04 |
| AWP | 66.05 ± 0.00 | 59.56 ± 0.00 | 69.05 ± 0.00 | 57.51 ± 0.00 |
| MCGC | 53.95 ± 0.00 | 61.78 ± 0.00 | 54.10 ± 0.00 | 57.25 ± 0.00 |
| mPAC | 56.80 ± 0.00 | 50.09 ± 0.00 | 60.10 ± 0.00 | 46.87 ± 0.00 |
| CGD | 98.60 ± 0.00 | 96.78 ± 0.00 | 98.60 ± 0.00 | 97.23 ± 0.00 |
| FPMVS | 78.45 ± 0.00 | 70.25 ± 0.00 | 78.45 ± 0.00 | 69.00 ± 0.00 |
| LMVSC | 83.70 ± 3.39 | 83.78 ± 1.20 | 91.93 ± 1.52 | 78.69 ± 2.84 |
| CoMSC | 98.50 ± 0.00 | 96.72 ± 0.00 | 98.50 ± 0.00 | 97.06 ± 0.00 |
| COMVSC | 61.15 ± 0.00 | 55.47 ± 0.00 | 61.35 ± 0.00 | 52.42 ± 0.00 |
| MMGC | 79.93 ± 0.03 | 65.24 ± 0.11 | 79.93 ± 0.03 | 65.90 ± 0.05 |
| SLMVGC | 99.25 ± 0.00 | 98.15 ± 0.00 | 99.25 ± 0.00 | 98.50 ± 0.00 |
| Ours | **99.55 ± 0.00** | **98.92 ± 0.00** | **99.55 ± 0.00** | **99.10 ± 0.00** |

**Table 5: The clustering results on ORL dataset (%)**

| Method | ACC | NMI | Purity | F-score |
|--------|-----|-----|--------|---------|
| Co-reg | 62.50 ± 0.94 | 80.17 ± 0.24 | 66.67 ± 0.62 | 52.37 ± 1.05 |
| Co-train | 57.42 ± 2.38 | 77.01 ± 1.32 | 60.92 ± 1.90 | 46.28 ± 2.14 |
| AWP | 77.25 ± 0.00 | 88.60 ± 0.00 | 80.00 ± 0.00 | 71.73 ± 0.00 |
| MCGC | 77.00 ± 0.00 | 87.22 ± 0.00 | 82.75 ± 0.00 | 56.25 ± 0.00 |
| mPAC | 63.75 ± 0.00 | 82.66 ± 0.00 | 66.75 ± 0.00 | 58.29 ± 0.00 |
| CGD | 54.25 ± 1.22 | 71.49 ± 0.90 | 61.17 ± 0.96 | 27.82 ± 2.30 |
| FPMVS | 55.50 ± 0.00 | 73.59 ± 0.00 | 59.00 ± 0.00 | 41.03 ± 0.00 |
| LMVSC | 64.67 ± 2.08 | 81.71 ± 1.16 | 73.67 ± 1.65 | 54.33 ± 1.82 |
| CoMSC | 72.75 ± 0.00 | 84.23 ± 0.00 | 77.25 ± 0.00 | 61.43 ± 0.00 |
| COMVSC | 76.50 ± 0.00 | 88.04 ± 0.00 | 81.50 ± 0.00 | 69.75 ± 0.00 |
| MMGC | 83.75 ± 1.80 | 90.82 ± 0.83 | 84.92 ± 1.77 | 77.06 ± 2.10 |
| Ours | **86.50 ± 0.00** | **93.32 ± 0.00** | **88.75 ± 0.00** | **82.02 ± 0.00** |

**Table 6: The clustering results on cora dataset (%)**

| Method | ACC | NMI | Purity | F-score |
|--------|-----|-----|--------|---------|
| Co-reg | 34.95 ± 2.01 | 19.68 ± 0.18 | 43.13 ± 0.39 | 26.94 ± 1.24 |
| Co-train | 53.31 ± 0.15 | 33.14 ± 0.72 | 58.30 ± 1.41 | 38.12 ± 0.19 |
| MLAN | 45.77 ± 0.02 | 24.45 ± 0.03 | 49.40 ± 0.02 | 34.34 ± 0.01 |
| AWP | 30.76 ± 0.00 | 14.27 ± 0.00 | 38.96 ± 0.00 | 28.17 ± 0.00 |
| MCGC | 32.05 ± 0.00 | 3.91 ± 0.00 | 32.20 ± 0.00 | 29.64 ± 0.00 |
| mPAC | 38.85 ± 0.00 | 20.08 ± 0.00 | 44.65 ± 0.00 | 29.11 ± 0.00 |
| CGD | 31.31 ± 0.00 | 2.75 ± 0.00 | 32.16 ± 0.00 | 30.71 ± 0.00 |
| FPMVS | 64.84 ± 0.00 | 40.23 ± 0.00 | 64.84 ± 0.00 | 45.09 ± 0.00 |
| LMVSC | 45.46 ± 3.84 | 27.47 ± 1.36 | 51.78 ± 1.92 | 32.51 ± 1.33 |
| CoMSC | 64.11 ± 0.00 | 46.47 ± 0.00 | 68.61 ± 0.00 | 49.36 ± 0.00 |
| COMVSC | 34.19 ± 0.00 | 11.04 ± 0.00 | 37.33 ± 0.00 | 30.33 ± 0.00 |
| MMGC | 56.68 ± 0.00 | 43.92 ± 0.00 | 65.84 ± 0.00 | 45.39 ± 0.00 |
| Ours | **69.04 ± 0.15** | **51.92 ± 0.10** | **70.24 ± 0.01** | **56.53 ± 0.14** |

**BBC**[1] dataset is composed of news stories in five different labels: politics, entertainment, business, tech and sport. **BBCSport**[2] contains 544 archives collected from the BBCSport website, where each document is divided into 2 kinds of features. **HW2**[3] is from

---

[1]http://mlg.ucd.ie/datasets/segment.html
[2]http://mlg.ucd.ie/datasets/segment.html
[3]https://archive.ics.uci.edu/dataset/72/multiple+features

the UCI repository.The dataset consists of 2000 samples with two views. Each sample is one of the handwritten digits (0–9). **ORL**[4] face dataset consists of 400 face images in 40 different themes in total. For each subject, the images are described in three features: facial expressions, facial details, and lighting. **Cora**[5] dataset consists of 2708 scientific publications classified into one of seven classes.

We selected 12 multi-view clustering methods for comparison: Multi-view Spectral clustering with Co-reg strategy (Co-reg) [10], Multi-view Spectral Clustering with Co-train strategy (Co-train) [9], Multi-view clustering and semi-supervised classification with adaptive neighbors (MLAN) [17], Multi-view Clustering via Adaptively Weighted Procrustes (AWP) [18], Multi-view Consensus Graph Clustering (MCGC) [32], Multiple Partitions Aligned Clustering (mPAC) [5], Multi-view Clustering via Cross-view Graph Diffusion (CGD) [24], Fast parameter-free multi-view subspace clustering (FPMVS) [28], Multi-view Subspace Clustering via Co-training (CoMSC) [13], Consensus One-step Multi-view Subspace Clustering (COMVSC) [33], Large-scale Multi-view Subspace Clustering (LMVSC) [7], Metric Multi-view Graph Clustering(MMGC) [22], Sample-Level Multi-View Graph Clustering(SLMVGC) [21]. Note that the above methods, except for the last one, perform graph fusion in a view-wise way. As for the last method, although it is instance-wise, it computes similarity graphs derived from topological manifold correlations, adept at capturing the entire topological manifold structure from the data space. In contrast, our approach using adaptive neighbor focuses more on the local geometric structure of data.

## 5.2 Results Analysis

Four criteria of clustering performance (Normalized Mutual Information (NMI), Accuracy (ACC), Purity, and F-Score) are shown in Table 1–6, and the best results are bolded. The comparison algorithms were repeatedly tested 10 times using the parameter settings recommended by corresponding papers. In most cases, our method consistently outperforms others, showing its effectiveness. In Table 1, on BBC dataset, our method outperforms other methods by at least 1.75%, 3.04%, 1.75%, 1.21% in terms of ACC, NMI, Purity, and F-score. For BBCSport dataset, our method achieves improvements around 3.86%, 2.86% and 0.55% respectively, while 2.25% is lower in F-score. Our method demonstrates the most significant advancement on BBC12 dataset in Table 3, and the corresponding improvements are 8.27%, 9.55%, 4.96%, 6.41%. Our method also yields competitive results for datasets where the majority of methods perform well, such as HW2 dataset. We can substantiate the effectiveness of mining the local geometric structure of similarity and integrating fine-grained information at the instance level by comparing empirical results. It should be pointed out that all the baselines, excluding the last one, suffer from a coarse-grained representation, resulting in suboptimal experimental outcomes. Regarding the last one, SLMVGC, our method demonstrates a discernible advantage in performance, which can be attributed to our empirical validation that local geometric structure captured by an adaptive neighbor is more tailored for fusion across views.

---

[4]https://www.kaggle.com/datasets/tavarez/the-orl-database-for-training-and-testing
[5]https://relational.fit.cvut.cz/dataset/CORA

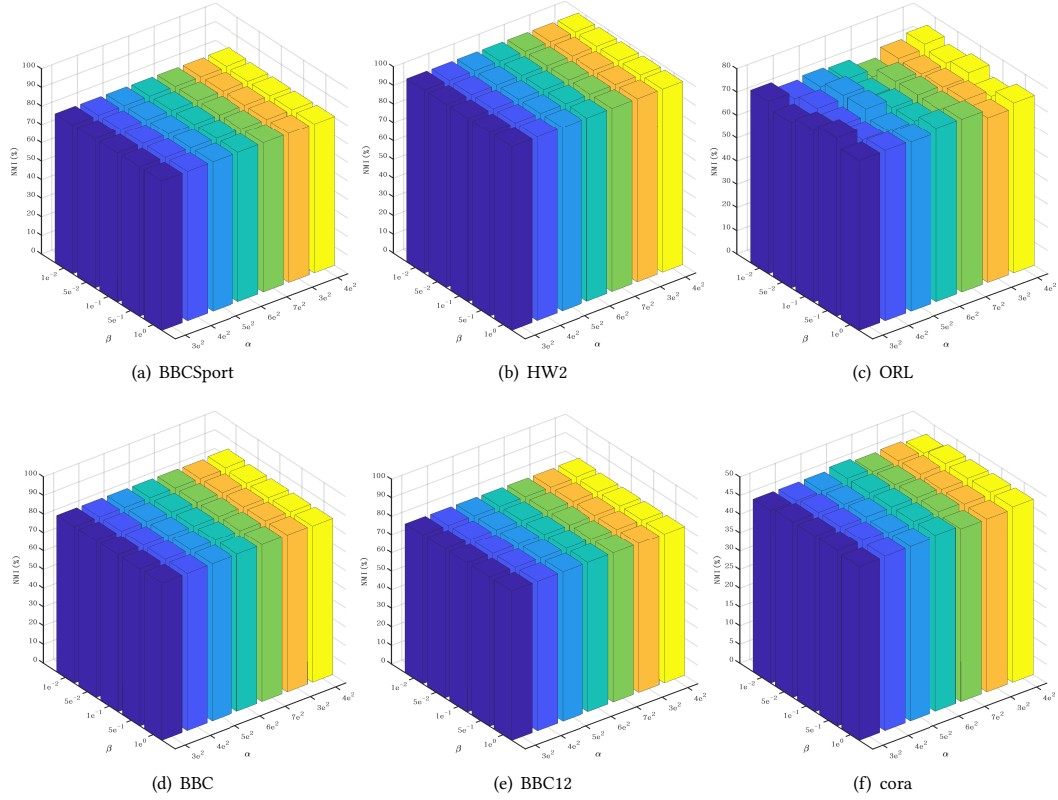

Figure 3: Parameter analysis

## 5.3 Parameter Analysis

There are four hyperparameters in our method: $\alpha, \beta, \rho,$ and $\lambda$. Note that $\rho$ is a common penalty parameter in ALM ($\rho = 1.2$), and $\lambda$ is fine-tuned according to Eq. (30). Therefore, we only need to focus on the remaining two parameters. To assess the impact of different parameter settings on clustering results, we vary parameter $\alpha$ and $\beta$ in $\left[3e^2, 4e^2, 5e^2, 6e^2, 7e^2\right]$ and $\left[1e^{-2}, 5e^{-2}, 1e^{-1}, 5e^{-1}, 1e^0, 5e^0, 1e^1\right]$, respectively. Figure 3 shows the visual results in five benchmarks. According to the experiment result, we conclude that our method demonstrates stability across a large range of $\beta$ settings and is robust in the case of a small value for $\alpha$.

## 5.4 Convergence Analysis

In this subsection, we experimentally verify the convergence of the proposed algorithm by reporting the corresponding loss value with the varying iterations on BBCSport and HW2. As shown in Figure 4, one may observe that the flexibility exists to seek the optimal solution for each variable, leading to the eventual convergence of the algorithm to a local minimum within 25 iterations. The proposed optimization algorithm is very efficient and converges fast. Hence it is sufficient for all datasets to reach the best performance with the maximum iteration numbers of 25, as set in our experiment.

## 5.5 Ablation Experiment

To validate that fusing instance-wise slices can better capture the local consistency, we directly apply spectral clustering to the similarity matrix of the individual views of neighboring instances, i.e., spectral clustering is performed on each $\{S^{(v)}\}_{v=1}^{m}$ learned by Eq. 1, where $m$ is the number of views. As shown in Table 5, similarity matrices learned on independent views with adaptive neighbor strategy are unreliable inputs for spectral clustering. MLAN, characterized as a view-wise fusion method that utilizes an adaptive neighbor strategy, demonstrates enhanced performance relative to methods employing a single-view approach. Furthermore, instance-wise fusion contributes to the most significant improvement of multi-view clustering outcomes by offering fine-grained representations, thereby augmenting local consistency.

## 6 CONCLUSION

In this paper, we propose exploiting local structure consistency across multiple views and focusing on mining fine-grained representations. The cross-view consistency in our model can also be captured by exploring the intersections of multiple views in an instance-wise manner. By utilizing the augmented Lagrangian method, our collaborative model can iteratively refine all subtasks towards optimal solutions. The empirical evaluation on various multi-view datasets shows that our method consistently outperforms other SOTAs in the majority of cases. Our method further

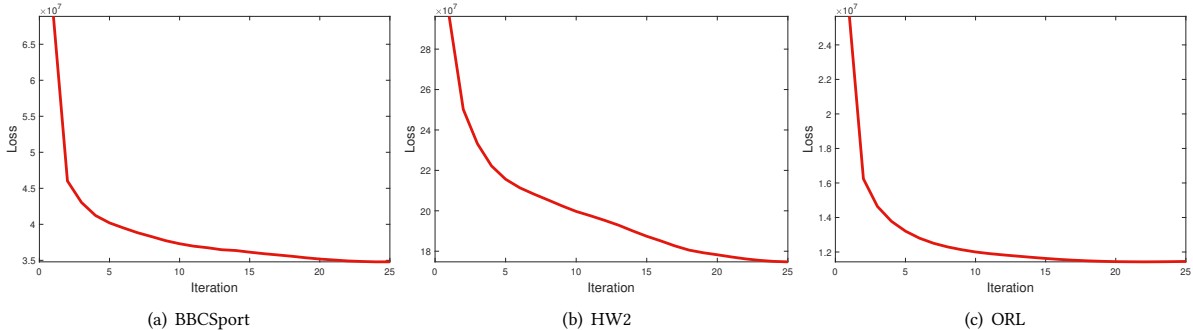

Figure 4: Loss convergence curve

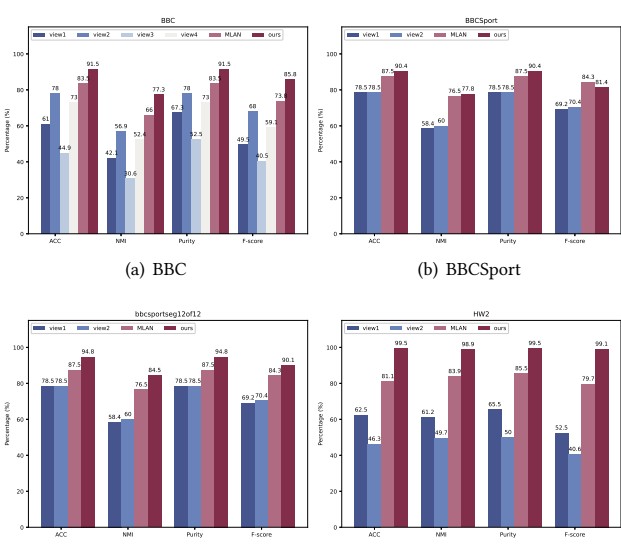

Figure 5: Ablation study

demonstrates stability across a large range of parameter settings. Note that the proposed optimization algorithm is very efficient and converges fast. Our work not only contributes to the ongoing exploration of multi-view clustering but also highlights the importance of considering local structure and the significance of fine-grained information fusion. In the future, we intend to extend the proposed model to other multi-view clustering frameworks such as subspace clustering and multi-kernel learning.

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
