# OpenReview forum: "Adaptive Instance-wise Multi-view Clustering"
_acmmm.org/ACMMM/2024/Conference — MM2024 Poster_

### Official Review · Reviewer_cvfv · 2024-04-30

**Rating:** 4
**Confidence:** 3

**Summary:**

This paper proposes an adaptive instance-wise multi-view clustering method which aims to exploit local structure consistency across multiple views and recover the fine-grained representations. Extensive experiments on several multi-view datasets confirm the significant enhancement in clustering accuracy achieved by our method.

**Strengths:**

(a) A novel method for instance-wise MVC by utilizing adaptive local structure is proposed in this paper, and experiments demonstrate that the proposed approach significantly improves clustering accuracy.
(b) The ablation experiments are conducted to verify the effectiveness of each module.

**Limitations:**

(a) Note that there are several paper have been proposed to explore the adaptive local structure, then what's the difference between this paper and other papers, in terms of the adaptive local structure?
(b) The dataset Cora, instead of cora, would be better.

**Suitability:**

2

---

### Official Review · Reviewer_G42n · 2024-05-23

**Rating:** 5
**Confidence:** 3

**Summary:**

This paper focuses on local structure consistency and fine-grained representations across multiple views with the help of augmented Lagrangian method. Experiments on several multiview datasets demonstrate that our proposed approach significantly improves clustering accuracy.

**Strengths:**

1. The method considers the local geometric structure and obtains view-specific similarity graphs using adaptive neighbors. This enables capturing detailed information within each view.

2. The method slices and fuses the multi-view similarity tensor at the instance level. By exploring the intersections of multiple views, it captures cross-view consistency, improving clustering performance.

3. The method employs the augmented Lagrangian method within a collaborative framework. This iterative approach refines subtasks towards optimal solutions, enhancing the accuracy of the clustering process.

**Limitations:**

1. The ablation study is conducted by comparing the MLAN with the proposed method, why is that? The essential difference between MLAN and the proposed method should be pointed out clear.

2. The overall presentation of the paper requires improvement. Clarity and coherence should be enhanced to ensure the reader can follow the content smoothly.

3. Some statements in the paper are confusing, particularly in the algorithm flow section (lines 365-374 on Page 4). It is necessary to clarify these statements to avoid any ambiguity or misinterpretation.

4. Given the abundance of notations used in the paper, including a notation table would greatly assist readers in better comprehending the content.

**Suitability:**

3

---

### Official Review · Reviewer_rrAm · 2024-05-24

**Rating:** 5
**Confidence:** 3

**Summary:**

In this paper, the authors perform fine-grained information fusion in a self-weighted manner by instance-wise slicing of the multi-view similarity tensor. A novel model that learns both the adaptive similarity matrix and the instance-level structure of multi-view data is introduced to obtain better clustering results. By skillfully combining these two subtasks, alternatively driving each subtask toward the optimal solution is obtained with the augmented Lagrangian method.

**Strengths:**

It is well motivated, the adaptive similarity matrix and the instance-level structure of multi-view data are well exploited simultaneously, which is impressive.

The experiments are sufficient and convincing. The comparison results demonstrate the effectiveness of the instance-wise learning strategy.

The paper is technically sound and overall well-written.

**Limitations:**

From Fig.1, I’m wondering what the essential differences between multi-view and multi-modal are? It seems they describe the same situation in some cases.

The results of CoMSC are worse than this work except F-score, why is that? It needs more detailed explanation.

The font size in Figure 2 is a little small.

**Suitability:**

3

---

### Official Review · Reviewer_sx9Q · 2024-05-24

**Rating:** 5
**Confidence:** 4

**Summary:**

This paper exploits the fine-grained consistency information by exploring the intersections of multiple views in an instance-wise manner, which contains the ideas on local similarity graph, multi-view similarity tensor, and slice fusion. Meanwhile, a collaborative framework with the augmented Lagrangian method is developed to refine all subtasks towards optimal solutions iteratively. Extensive experiments can show its effectiveness.

**Strengths:**

1. The paper introduces a novel approach to multi-view clustering by focusing on local structure consistency and fine-grained representations across multiple views.
2. The paper provides a solid theoretical foundation for its proposed method, which is mathematically sound and well-justified.
3. The method is evaluated extensively on several multi-view datasets.
4. The paper is well-organized and clearly written.

**Limitations:**

1. The paper compares the proposed method with a limited set of baseline methods, especially recent neural network-based multi-view works.
2. The paper does not provide a thorough evaluation of the scalability of the proposed method.

**Suitability:**

3

---

### Meta-Review · Area_Chair_2KBD · 2024-07-01

**Recommendation:** Accept (Poster)
**Confidence:** 5

**Metareview:**

This paper focus on local structure consistency and fine-grained representations across multiple views, which is theoretically reasonable and technically sound. The structure of the paper is clear, and the experiment has achieved good performance. All reviewers give positive scores and tend to accept this work.